# Qualitative classification of thrombus images as a way to improve quantitative analysis of thrombus formation in flow chamber assays

**Piotr Kamola, Tomasz Przygodzki** [ORCID] *

Department of Haemostatic Disorders, Medical University of Lodz, Lodz, Poland

* tomasz.przygodzki@umed.lodz.pl

## Abstract

### Background

Thrombus formation in vitro under flow conditions is one of the most widely used methods to study haemostasis and to evaluate the activity of potential antithrombotic compounds. Assessment of the results of these experiments is often based on a quantification of microscopic images of thrombi. In a majority of reported analysis all thrombi visualised in an image are quantified as one homogenous class. In some protocols, qualitative assessment of thrombi morphology based on a visual comparison of evaluated images with representative images of predefined classes of thrombi are performed by experienced analysts. In presented paper we show how the quantitative analysis can be improved by classification of thrombi on the basis of defined morphological features prior to quantification and we suggest that machine learning-based approach can improve this way of analysis.

### Methods

We tested the applicability of machine learning-based segmentation and classification of thrombi images to improve the outcome of quantification of the results of flow chamber assays. For this, we used the public domain machine learning software Ilastik for bioimage analysis developed at the European Molecular Biology Laboratory. A model was trained to distinguish two classes of thrombi based on certain morphological features which apparently correspond to the stage of thrombus development. Thrombi formed in the presence of a model antiplatelet compound—abciximab or in control conditions were quantified with the use of this model and the results were compared to quantification where all thrombi were quantified as a homogenous class.

### Results

Machine learning-based analysis was capable of effective distinguishing of two classes of morphologically distinct platelet aggregates. The use of the model which segmented and quantified only the objects recognized as compacted structures provided results which better mirrored the actual effect of an antiplatelet treatment than quantification based on all structures.

**Data Availability Statement:** A detailed description of both workflows is provided in Supplementary materials. All the steps of the presented workflows were automated by scripts written in python

programming language. The python scripts, Fiji macros, Ilastik trained models and raw data were deposited in Zenodo repository (https://doi.org/10.5281/zenodo.10495557) and are accessible for potential users.

**Funding:** This work was supported by the National Science Centre grant OPUS (UMO-2020/37/B/NZ3/00301).

## Conclusions

Classification of thrombi enabled by machine learning increases the relevance of quantitative information and allows better evaluation of the results of in vitro thrombosis assays.

## Introduction

Thrombus formation in vitro under flow conditions is one of the most widely used methods to study haemostasis and to evaluate the activity of potential antithrombotic compounds [1, 2]. In this method, blood is perfused through channels coated with protein components which stimulate the activation of blood platelets and the formation of thrombi. There are several ways of quantification of the effectiveness of thrombi formation in these systems. One of them is based on the assessment of the resistance of blood flow, which increases with the growth of occlusive thrombi [3]. A relatively novel approach uses quartz crystal microbalance system [4]. However, the most often used method of quantification is based on the evaluation of microscopic images of thrombi [5–7].

In a majority of reported analysis all visualised thrombi are quantified as a homogenous class. At the same time however, different classes of thrombi can be distinguished in a single image with respect to such features as their morphology or intensity of staining. This non-homogenous morphology of thrombi across the image results from different stage of their growth. In selected studies researchers used a score assessment of thrombi morphology to express the differences in their appearance in a quantitative way [6, 8, 9]. This approach was based on a visual comparison of evaluated images with representative images of predefined classes of thrombi which was performed by experienced analysts.

Presented work is aimed at providing a solution which would allow thrombi qualitative classification prior to quantification based on machine learning-based classifier rather than on the classification by the eye of an analyst. We focused on distinguishing thrombi which, in the moment of imaging, presented apparently different stage of development. The aim of the study was to train a classifier capable of semantic segmentation of only highly grown, compacted thrombi and to test if quantification of thrombi of solely this class would have advantage over quantification of all thrombi. Creating such a solution would be beneficial in the case of analysis of the effects of treatments which result in a decrease of height and volume of thrombi but which leave the total area covered by platelets unaffected. In these cases the effect of treatment can hardly be quantified on the basis of wide-field or single focal plane confocal images, which are the most often used imaging techniques. The evaluation of the effects of certain treatment on the formation of high, compacted thrombi is crucial as inhibition of this process is the actual aim of antiplatelet and antithrombotic therapies.

To validate the proposed approach, images of thrombi formed in the control conditions and in the presence of a model inhibitor of platelet aggregation–abciximab were quantified by our trained model and by a classical approach where all structures were taken into calculus.

In the field of bioimaging, there are currently several software solutions that allow performing machine learning–based analyses. One such solution is a public domain machine learning software for bioimage analysis Ilastik created at the European Molecular Biology Laboratory distributed under GNU General Public License which we decided to train as a classifier in our studies [10].

## Methods

### Generation and imaging of thrombi

The experiments were performed with the use of the VenaFlux platform (Cellix, Dublin, Ireland). The channels of the Vena8 Endo+ biochip were coated with type I collagen (Chrono-log Corp., Havertown, PA, USA) in concentration of 20 μg/ml or fibrinogen (Calbiochem, Darmstadt, Germany) in concentration of 100μg/ml overnight at 4˚C and blocked with 0.1% BSA for 1 hour at room temperature. Blood samples were collected from healthy volunteers (n = 15, age between 20–55 years, 10 female and 5 male, platelet count = 266±45 plt/μm$^3$ [mean±S.D.]) into a vacuum tube containing 0.105 M buffered sodium citrate. Blood was recalcified and perfused with shear force of 20 dynes/cm$^2$ (approx. 445s$^{-1}$) for 4 minutes. In some experiments, abciximab (Janssen Biologics B.V., Leiden, Netherlands) was added prior to perfusion at a final concentration of 10 μg/ml. Measurements were performed in a paired manner i.e. blood sample was split in two and one sample was added with abciximab and the other with equal volume of saline. Adherent platelets were stained with platelet-specific anti-CD41 PE-conjugated antibodies (Becton-Dickinson, San Diego, CA, USA). Imaging was performed using a wide-field epifluorescence AxioExaminer microscope (Carl Zeiss, Oberkochen, Germany). At least eight images were acquired in each channel. The study was approved by the Medical University of Lodz committee on the Ethics of Research in Human Experimentation (number of the consent RNN/323/20/KE). Volunteers presented informed consent. Volunteers were recruited between January 2022 and May 2023.

### Training of models in Ilastik

Two classes of thrombi were distinguished: one class comprised of objects relatively homogenously stained across their area and the second consisted of objects with strong border staining and relatively dark centre **Fig 1A.** These two classes of thrombi apparently represent structures on different stages of development as can be deduced by comparison with three-dimensional reconstruction of confocal images of thrombi generated in the same conditions **Fig 1B**.

The difference in staining presumably results from various permeability of the structures to fluoro-labelled antibodies. It could be assumed that the structures which are less grown in z-axis and plausibly less densely packed are relatively homogenously stained as a majority of platelets is accessible for antibodies. The second group consists of packed, built up thrombi with lower accessibility of antibodies to their inner region. For the sake of convenience the latter class of thrombi will be denominated in the text as compacted structures. We hypothesised that quantifying only the structures included in this group would provide more discriminatory results than the classical approach in which all CD41-positive structures are quantified.

In order to compare these two approaches, two models were trained in Ilastik. One segmented all of CD41 positive objects and the second capable of differentiating between two morphologically different classes of platelet aggregates.

The segmentation in both models was a two-step process. A first step, pixel classification, is aimed at selection of pixels which form objects of interest. In the second stage, object classification, the pixels selected at the first step are grouped into objects and these objects are further assigned to different classes. Training of both models was performed according to the following schemes:

i) Model for quantification of all objects irrespective of their morphology:

two classes of pixels were created: background and the area covered by platelets. In an object classification step all objects identified as platelet aggregates were classified as thrombi.

ii) Model for quantification of only compacted aggregates:

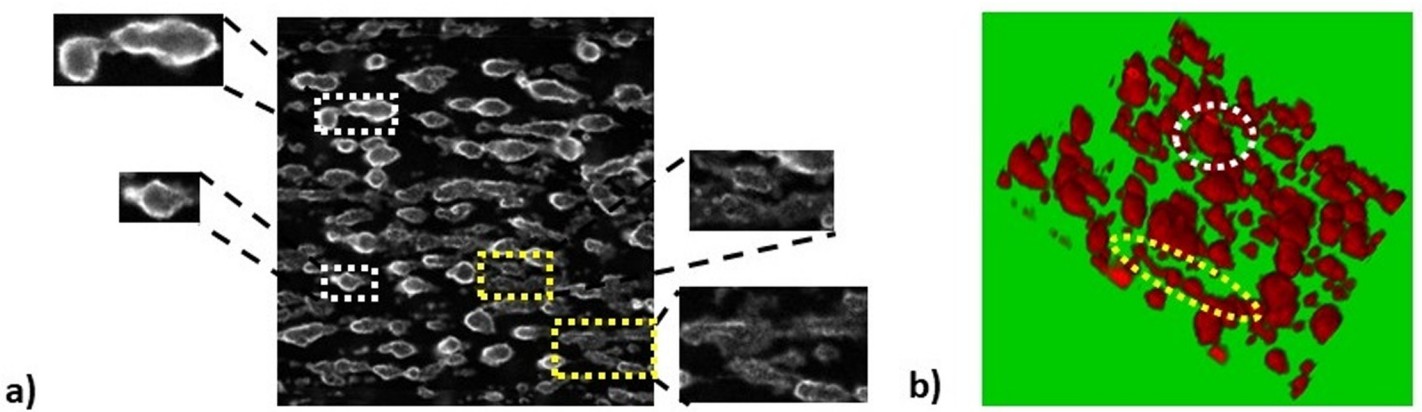

**Fig 1. Morphological heterogeneity of platelet aggregates formed on collagen under flow conditions.** a) an image of thrombi formed on a collagen-coated surface acquired by wide-field microscopy, inlet graphs represent examples of structures with strong border staining and relatively dark centre (white frames) and of homogenously stained structures (yellow frames). b) confocal image of thrombi formed on a collagen-coated surface, an example of thrombus spanning the height of a channel (white frame), an example of low thrombus (yellow frame).

In the pixel classification step two classes of pixels were defined: 1) background, 2) CD41 positive structures.

In the objects classification step the objects formed of the second class of pixels were taken into account. These objects were separated into two classes. It is relatively difficult to provide clear criteria to differentiate objects of such morphological complexity as platelet aggregates and thrombi. Proposed criteria of differentiation of these objects must therefore be to a large extent arbitrary. The criteria used to train our proposed model were the following: objects with strong edge staining on most of the perimeter and with relatively darker centre were classified as compacted thrombi, whereas objects where staining was more uniform, and the structures presumably less compacted, were classified to the second group. The schematic representation of this training procedure is presented on **Fig 2.**

Detailed description of both training procedures is provided in **S1 File**.

To train both models, images acquired in flow chamber assays on blood samples from 9 different donors were used. The model was trained on total 18 images. Total number of objects used to train the model which quantified only the compacted structures was 3750.

The analysis were performed with the use of 1.4.0b21 Ilastik version.

The workflow was run on Intel Core i7-10700K CPU equipped with 32GB DDR4 RAM under 64-bit Windows 10 Pro.

## Image analysis

The first step consisted of image preprocessing with the use of FIJI software [11]. At this step the file format was changed from the one generated by the microscope software, i.e. Zeiss .czi format, to tagged image file format (tiff). At this step the background substraction was also performed with the use of Substract Background function in FIJI. The entire preprocessing step was automated with the use of macro written in IJ1 Macro language.

In the next step, Ilastik models trained as described above were used to perform pixel classification followed by object classification to generate maps of objects for each analysed image. The model aimed at segmentation of all objects generated the maps of all structures consisting of blood platelets. The second model in turn generated maps of probability that a certain structure belongs to the class of compacted thrombi.

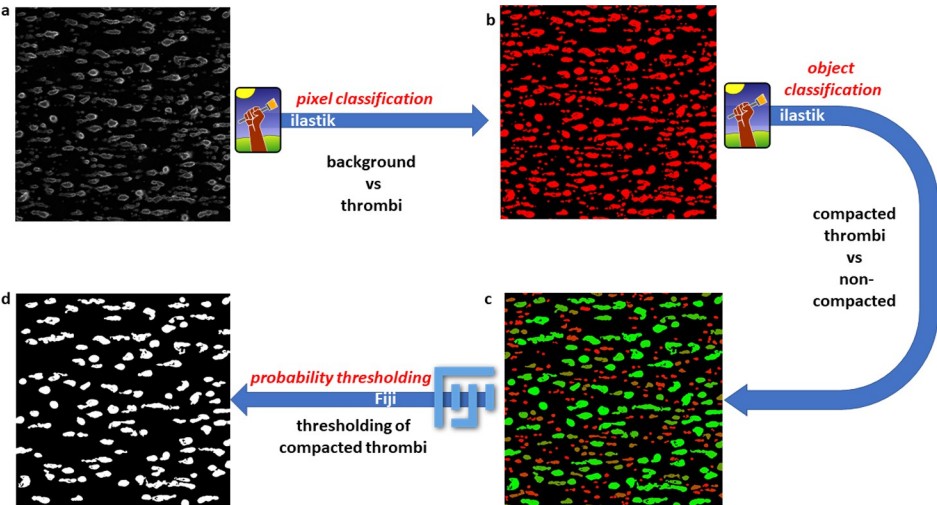

**Fig 2. Schematic representation of pixel and object classification steps in Ilastik followed by particle analyses by FIJI in the workflow where only selected class of thrombi was quantified.** a) an exemplary raw image of CD41-labeled blood platelets aggregated on collagen-coated surface; b) pixel classification map: red spots represent groups of pixels which with high probability belong to the class of platelets; c) Object prediction map where two classes of objects were defined: compacted thrombi and non-compacted aggregates. Colours represent probability that object belongs to the compacted thrombi class: green colour for high probability and red for low probability. d) object probability map thresholded to contain only the objects which belong to the compacted thrombi class with certain probability.

The objects on the maps were then quantified by Analyze Particles module of FIJI. In the maps generated by the first model all objects were quantified. The maps generated by the second model were thresholded at first place, so that only the structures which belonged to the class of compacted thrombi with a probability higher than the threshold were further processed. The choice of this value was determined during model evaluation described in Results section.

Parameters calculated by the Analyze Particles tool in FIJI included area of individual thrombi and Feret diameter of each thrombus. "Include holes" option was checked so that even if a segmentation was empty inside, the values were calculated for the entire segmented area. The operation was performed with the use of macro written in IJ1 Macro language.

Finally python script was run to calculate mean and median values of the parameters and summarized area of the structures segmented in each image.

Schematic presentation of the workflow is shown in **Fig 3.** All the steps of the workflow were automated by a single python script which was launching in a sequential manner all of the workflow components: FIJI macro for preprocessing, python script for Ilastik pixel prediction followed by object prediction in headless mode, FIJI macro for particle analyses, and finally python script to generate excel files.

## Statistical analysis

Normality of data distribution or of differences between paired values was tested with The Shapiro–Wilk test and Levene's test. The statistical significance of differences between two groups was estimated using the paired or the unpaired Student's *t*-test. To compare differences between more than two groups ANOVA for repeated measurements and the Tukey's *post-hoc* test for multiple comparisons were used. Analyses were performed and charts were generated with use of GraphPad Prism v.9 (San Diego, CA, USA).

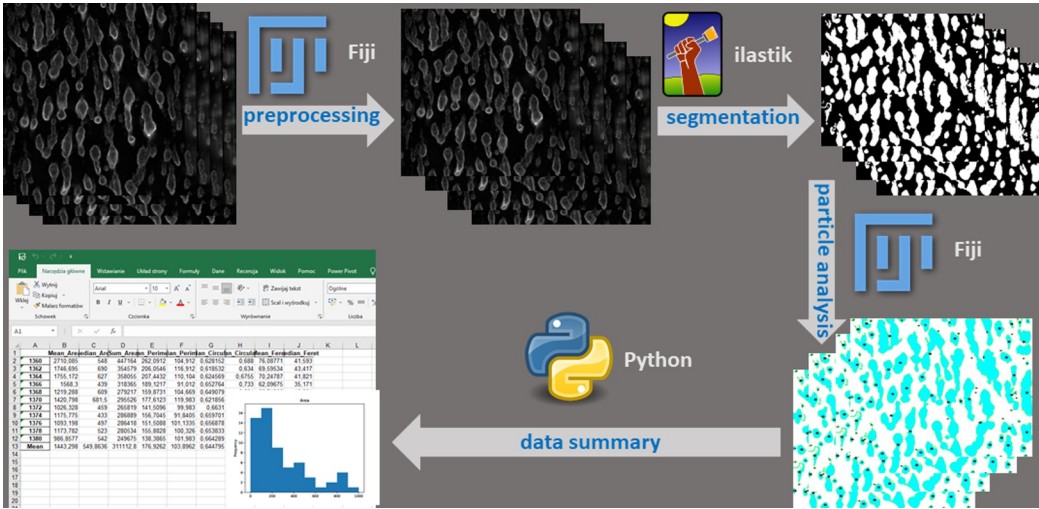

**Fig 3. Schematic representation of the workflow used for image quantification.** The raw images are first pre-processed with the use of Fiji software. In a next step Ilastik model is used to segment the structures and probability maps are generated for each image. The maps are analysed with the use of Fiji to provide quantification of the structures. Finally Python script is used to generate spread sheets which contain measurement results.

## Ethics approval and consent to participate

The study was approved by the Medical University of Lodz committee on the Ethics of Research in Human Experimentation (number of the consent RNN/323/20/KE). Written informed consent, including detailed information regarding the study objectives, study design, risks and benefits, was obtained from each individual before blood withdrawal.

# Results

## Evaluation of the model

The evaluation of the model used to segment all aggregates was based on comparison of consistency of model-driven segmentation with that performed by an analyst. For this purpose, 10 cropped fragments of images were used. Ground truth segmentation was obtained by manual outlining of the objects by an analyst. The same cropped fragments were segmented by the trained model. Both segmentations were overlapped using Python script which calculated ratio of intersection of the objects (area which is shared by overlapping objects) and union (summarized area of overlapping objects) for each image and intersection over union metric (Jaccard index) was calculated. Jaccard index calculated for the model which segmented all thrombi equalled to 0.89 (0.86; 0.92) (median; IQR). The index values higher than 0.7 are generally considered indicative of high coverage.

In the second model, its ability to differentiate between compacted and non-compacted thrombi was assessed. In a set of images, outlines of all structures segmented by the model were visualised and assigned by an analyst to either of the classes to generate ground truth classification. In the next step, the assignments were compared with classification of structures as predicted by the model. As described above, Ilastik generated probability maps where each structure was assigned with a probability of being a solid thrombus. These probability values were than used to threshold the structures in the last step of the workflow: the structures assigned with the probability value higher than the threshold value were considered as thrombi and further processed. The selection of this threshold value was performed in a procedure

described below. The process of comparing ground truth with model predictions was performed for 22 different threshold values of probability ranging from 0.16 to 0.98. To facilitate the visual inspection of the images the probability values were recalculated from 0–1 range to 0–255 (8-bit image). In the following sections of the manuscript these renormalised values will be used to express thresholds.

The outcome of this comparison was summarized for each image and for each threshold value in a form of a confusion matrix. When a structure labelled by an analyst as compacted thrombus was predicted accordingly by the model it was considered a true positive (TP), and if the model recognised it as non-compacted, it was considered a false negative (FN). In turn, when a structure labelled by an analyst as non-compacted was predicted accordingly by the model it was considered a true negative (TN), and if predicted by the model as compacted it was considered as false positive (FP). The values were then used to calculate true positive rate (TPR)–a fraction of all compacted thrombi correctly identified as such and false positive rate (FPR)–a fraction of non-compacted structures mistakenly predicted by the model as compacted. These metrics calculated for each threshold value were plotted against each other to generate ROC curves for each image separately and from them an averaged ROC curve was generated (Fig 4). Such presentation of metrics is widely used to evaluate the quality of tests. It shows a balance between positive samples that are detected at a given threshold value and negative samples which are falsely detected as positive in the same conditions. The procedure was performed with use of ImageJ macros and python scripts. The detailed description is provided in S1 File.

Two threshold values were chosen for further analysis: 120 and 150. At the threshold of 120 a TPR value equalled to 0.98, and FPR equalled to 0.27 meaning that only approx. 2% of compacted structures were missed in the analysis, but this high recall was at the expense of approx. 27% of non-compacted structures being falsely identified as compacted. In the case of threshold equal to 150 TPR equalled to 0.78, and FPR equalled to 0.15 meaning that up to approx. 22% of compacted structures were missed in the analysis, but the rate of non-compacted

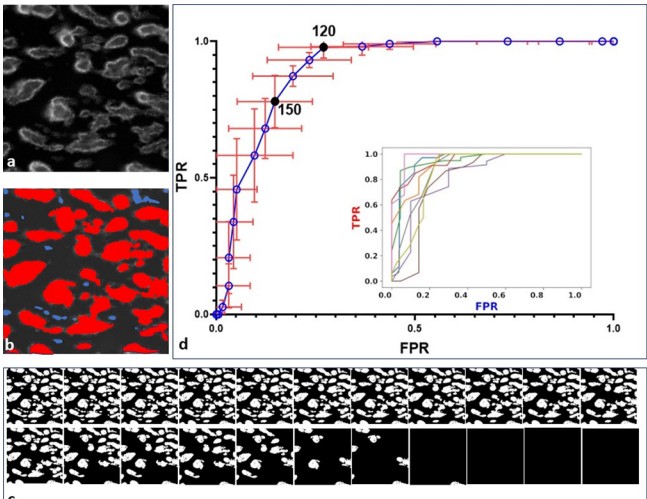

**Fig 4. ROC curve for the model used to segment only compacted thrombi.** a) an exemplary raw image of CD41-labeled blood platelets aggregated on collagen-coated surface; b) schematic representation of ground truth labelling: blue selections–non-compacted structures, red selections–compacted structures; c) the exemplary image thresholded at 22 different values of probability d) ROC curve—true positive rate (TPR) and false positive rate (FPR) values were calculated on the basis of confusion matrices generated by comparison of binary images thresholded at various values of probability with ground truth. Inlet graph shows individual ROC curves for each of nine images used to calculated confusion matrices.

structures falsely identified as compacted dropped to approx. 15%. A further increase of the threshold to minimise FPR would result in a substantial decrease in thrombi detection as can be read out from the ROC curve.

## Quantification of thrombi with the use of the trained Ilastik models

At the first place the model was tested to assess its ability to distinguish thrombi from layers of adhered platelets which do not form structures substantially developed in z-axis. To this end we used images of platelets adhered on fibrinogen. In these conditions blood platelets tend to coat the surface with relatively thin layer and do not form thrombi as it occurs on collagen coated surface. The images used at this step were not included in the training set.

In the case of nearly confluent coverage of the surface with platelets, the model segmented and quantified only a fraction of objects where platelets were specifically densely packed. This resulted in a significantly lower values of coverage area at both thresholds when compared to analysis with the classical approach where all CD41-positive structures were segmented (**Fig 5**). This confirmed the specificity of the model towards such structures.

As the next step we used the model to quantify the effects of a model antiplatelet drug and to compare the outcome of the quantification with the most often used approach based on quantification of all platelets adhered to the surface. Abciximab caused an apparent decrease in the number of aggregates and many of them did not present a morphology which fulfilled the criteria of compacted structures (**Fig 6**).

When the objects were segmented without respect to their morphology, the area covered by platelets did not significantly differ between control and abciximab samples (**Fig 7A**). However, when instead of the classic approach, a model was used for segmentation and only the high aggregates were quantified, the area covered by platelets was significantly smaller in abciximab treated samples at both thresholds applied (**Fig 7B and 7C**).

Inhibition of platelet aggregation may lead to formation of a number of smaller thrombi instead of lower number of larger structures. In such case a total coverage area is not a relevant

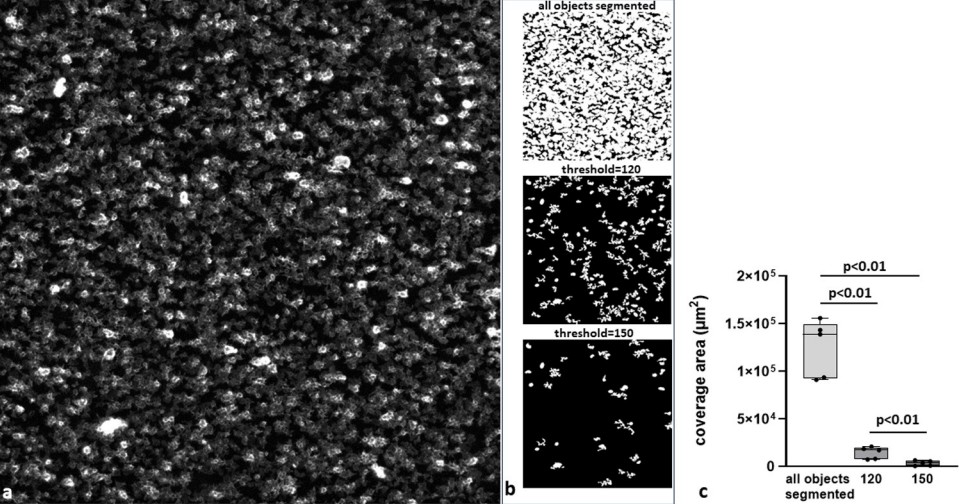

**Fig 5. Quantification of platelets adhered to fibrinogen.** a) an exemplary raw image of CD41-labeled blood platelets adhered to fibrinogen-coated surface; b) the exemplary image segmented with the model where all CD41-positive objects are segmented (upper image), with the model trained to segment only the compacted thrombi thresholded at 120 (middle image) and 150 (bottom image); c) coverage area calculated for three ways of segmentation depicted in b); ANOVA for repeated measurements and the *post-hoc* test for multiple comparisons, n = 5, p-values two-tailed.

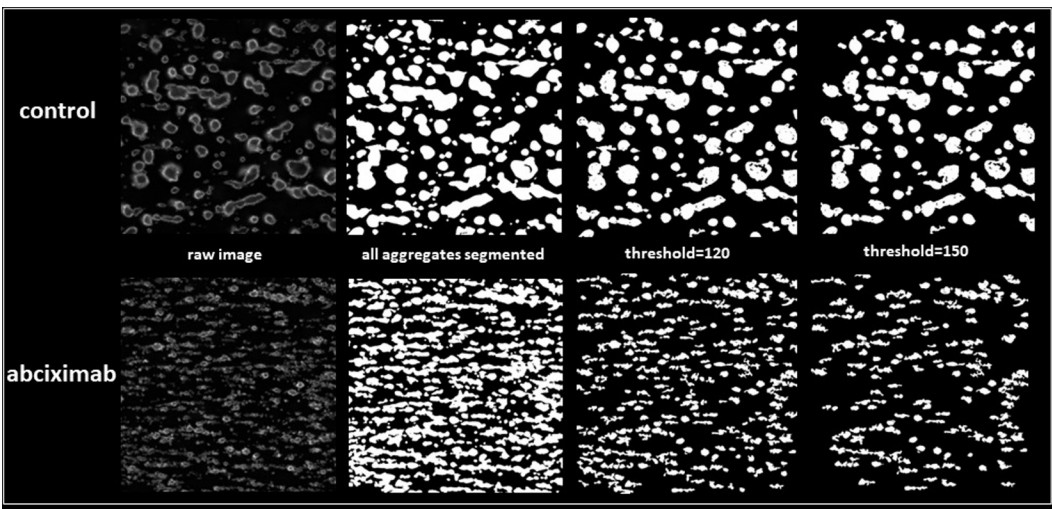

**Fig 6. Segmentation of thrombi formed in control conditions and in the presence of abciximab with the use of the models described in the study.** Exemplary images of platelet aggregates generated on collagen-coated surface in control conditions (upper panel) and in the presence of abciximab (lower panel). Raw images are shown along with the binary maps of segmentations of all structures and with the use of the model which detects only the compacted thrombi thresholded at the values of 120 and 150.

measure of the effect and a size of individual thrombi is used as a parameter of choice. In presented results abciximab treatment resulted in a significant decrease of a median size of individual thrombi even when they were quantified as a homogenous class (**Fig 7D**). The difference however was much more pronounced when only solid structures were segmented with the use of the model (**Fig 7E and 7F**).

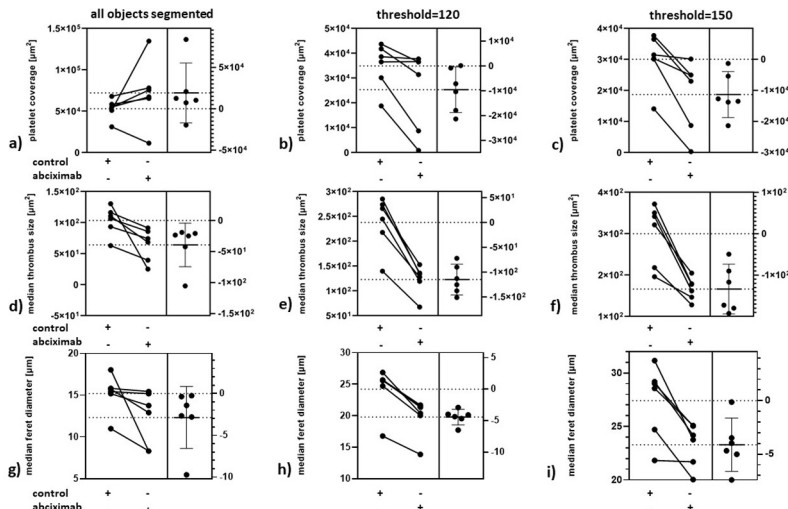

**Fig 7. Quantification of the effects of abciximab on thrombi formation with the use of the models described in the study.** Area covered by platelets based on segmentation of all platelet aggregates (n.s.) (a) and based on segmentation of only compacted aggregates thresholded at 120 (p = 0,0436) (b) or at 150 (p = 0,0108) (c). Median area of individual aggregates based on segmentation of all platelet aggregates (p = 0,0348) (d); and based on segmentation of only compacted aggregates thresholded at 120 (p = 0,0002)(e) or at 150 (p = 0,0023)(f); Feret diameter of individual aggregates based on segmentation of all platelet aggregates (n.s.) (g) and based on segmentation of only compacted aggregates thresholded at 120 (p = 0,0003) (h) or at 150 (p = 0,0082) (i); n = 6, all analyses performed with paired t-test, all p-values two-tailed, each point represents a mean of value calculated for at least eight images.

Another parameter which describes the morphology of individual objects is Feret diameter. As it is calculated by FIJI, the value represents the longest distance between any two points along the selection boundary. In the case of thrombi, this parameter would therefore be a measure of the ability of the haemostatic system to form continuous structures under the shear stress. When all objects were quantified Feret diameter did not differ between control and abciximab-treated samples (**Fig 7G**). It was however significantly decreased in abciximab samples when only compacted thrombi were quantified (**Fig 7H and 7I**).

Segmentation of the selected class of structures resulted not only in an increase of the difference between control and abciximab-treated samples, but also in a change in the absolute values of each parameter. The coverage area in both control and abciximab samples decreased with the application of the model selective towards the compacted structures, which reflected the fact that less structures were segmented and quantified. On the other hand median area and median Feret diameters increased when the model was applied. This was an effect of exclusion of the smaller fraction of objects from the analysis.

## Discussion

In many of the experiments conducted with the use of flow chambers quantification is based on images acquired by means of wide field microscopy or confocal imaging of a single focal plane. In such analysis all thrombi are considered as a homogenous class and quantitative representation of images includes total area covered by these structures or a central measure tendency of certain parameters of individual thrombi. In presented work we suggest another approach where quantification is based on selected structures which fulfil a certain morphological criterion. In our particular example this morphological type corresponded to the stage of thrombi growth.

Quantification with the use of a model trained to segment only the structures fulfilling these criteria assured better discrimination of the inhibitory effect of the model antiplatelet drug than quantification of all structures. In the latter case evaluation of parameters of individual objects provided a certain discrimination which might suggest that this solution is sufficient to achieve the aim and that qualitative selection of structures prior to quantification does not provide additional benefits. In fact, when a strong inhibitory effect is considered such a solution may provide satisfactory results. In the case of more subtle effects however, qualitative selection may be useful. Also, if an inhibitory effect on thrombus formation was accompanied by a substantial increase of adhesion of platelets leading to coating of the surface, quantification of individual structures treated as one class would become more difficult. In such case, as shown by the example of platelets adhered to fibrinogen, the trained model capable of distinguishing layers of platelets from more compacted aggregates may prove useful.

In the life sciences there exist no analytical methods that provide indisputable results. Quantification of images based on machine learning model is no exception. The main source of variability in this case is uncertainty about the classification of the objects. To verify how different classification can affect the results we compared the outcome of analyses at two different values of threshold. The comparison showed that even when approximately 20% of structures not fulfilling our criteria were taken into calculus at low threshold or when approximately 20% of the structures of interest dropped out from analyses at high threshold, still the model more efficiently discriminated the effect of antiplatelet treatment than analyses based on all objects.

Presented method of analyses is by no means intended to serve as a replacement for confocal microscopy with three-dimensional reconstruction of images. The latter will always provide the most relevant quantitative details such as volume or height of thrombi. At the same time however, wide-field imaging or confocal imaging of a single focal plane remain the most

often ways to record thrombi images. It is due to several reasons. Acquiring of z-stack images with the use of confocal microscopy is relatively time-consuming and laborious task. This is of particular significance in the case of flow chamber assays where several images along the channel should be acquired to provide a relevant representation of thrombi heterogeneity. Additional factor which limits the usage of confocal microscopy is relatively lower accessibility to such setups when compared to wide field microscopes. Therefore we believe that solution which improves a relevance of quantitation based on single focal planes images may prove beneficial.

The idea of quantification combined with qualitative distinction between morphological classes of thrombi has been already applied in several studies [6, 8, 9]. It has been based on comparing by an analyst of images acquired in the experiments with reference images representative to the classes of interest. The images were then assigned a score value depending on the outcome of the comparison. Machine learning approach proposed in the paper could support this way of analysis. The model can be trained to distinguish thrombi not only on the basis of their morphology, like in the presented paper, but also on the basis of staining of specific markers of platelet activation. Such an approach has been recently used to distinguish platelet aggregates varying in terms of the density of GPVI clusters [12]. The same solution has proved useful for classifying of adhered platelets with respect to different degree of spreading [13], or to detect platelets characterized by compromised spreading in a cohort of patients with bleeding disorders [14].

All the steps of the presented workflows were automated by scripts written in Python programming language. An automated workflow which combines segmentation with quantitative analysis allows fast calculation of experimental data from a set of images. This is of special importance when re-evaluation of a large number of images is required. The latter scenario may occur when novel experimental data or literature survey inclines an analyst to look at the already analysed data from a different perspective. For instance, if such a novel protocol of analysis demanded an alternative object classification, this could be easily achieved by training of a new model and replacing the respective files in the workflow. Similarly, a calculation of a novel quantitative parameter describing the objects requires only a modification of a selected script file. The workload of re-analysis is therefore minimized and its time is limited only by a computational power of a computer setup. Alternatively such workflows can be created with the use of graphical environments such as KNIME [13].

## Conclusions

In presented work we propose a machine learning based solution to segment thrombi in fluorescently labelled images in the process of quantification of experiments performed in flow chambers. We suggest that the use of classification of thrombi enabled by machine learning may increase the relevance of quantitative information and allow better evaluation of the results of these experiments.

## Supporting information

**S1 File.**
(DOCX)

## Author Contributions

**Conceptualization:** Tomasz Przygodzki.

**Data curation:** Tomasz Przygodzki.

**Formal analysis:** Tomasz Przygodzki.

**Funding acquisition:** Tomasz Przygodzki.

**Investigation:** Piotr Kamola, Tomasz Przygodzki.

**Methodology:** Tomasz Przygodzki.

**Project administration:** Tomasz Przygodzki.

**Resources:** Tomasz Przygodzki.

**Software:** Tomasz Przygodzki.

**Supervision:** Tomasz Przygodzki.

**Validation:** Tomasz Przygodzki.

**Visualization:** Tomasz Przygodzki.

**Writing – original draft:** Tomasz Przygodzki.

**Writing – review & editing:** Piotr Kamola, Tomasz Przygodzki.

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
