## [Decision Letter · Decision Letter 0]

28 Nov 2023

PONE-D-23-31983Qualitative classification of images of thrombi as a way to improve quantitative analysis of thrombus formation in flow chamber assaysPLOS ONE

Dear Dr. Przygodzki,

Thank you for submitting your manuscript to PLOS ONE. After careful consideration, we feel that it has merit but does not fully meet PLOS ONE’s publication criteria as it currently stands. Therefore, we invite you to submit a revised version of the manuscript that addresses the points raised during the review process.

The manuscript focuses on a topic of potential interest. The study, however, presents major pitfalls that should be addressed to support sound conclusions. To mention some of them, ii) lack of information on donor variables, including platelet count and sex; ii) lack of comparison to a manual analysis and verification for validation of the machine learning approach; iii) lack of clarity on the choice of ‘only high aggregates’; iv) analyse images across a series of timepoints to allow the identification of individual thrombi once the platelet coverages becomes confluent; v) provide more details on the experimental setting, such as figure legends, number of the experiments performed, how many images were analysed for each perfusion, statistical analysis, set up of the experiments with and without abciximab.

We look forward to receiving your revised manuscript.

Kind regards,

Giuseppe Remuzzi

Academic Editor

PLOS ONE

Journal Requirements:

"This work was supported by the National Science Centre grant OPUS (UMO-2020/37/B/NZ3/00301)."

Reviewers' comments:

Reviewer's Responses to Questions

**Comments to the Author**

1. Is the manuscript technically sound, and do the data support the conclusions?

Reviewer #1: Partly

Reviewer #2: No

2. Has the statistical analysis been performed appropriately and rigorously? 

Reviewer #1: No

Reviewer #2: No

3. Have the authors made all data underlying the findings in their manuscript fully available?

Reviewer #1: Yes

Reviewer #2: No

4. Is the manuscript presented in an intelligible fashion and written in standard English?

Reviewer #1: No

Reviewer #2: Yes

5. Review Comments to the Author

Reviewer #1: The work of Kamola and Przygodzki is original and interesting. They used machine learning-based analysis to improve classification of platelet thrombi that results from flow chamber assays. The methodology described could turn useful for fast re-analyses of thrombus formation. However, the work is too preliminary, and some points should be added or improved as follows:

- As correctly stated by the Authors the methodology described should not be intended to serve as a replacement of confocal microscopy with three-dimensional reconstruction of images. The new method can help implement image re-analysis as it enables fast calculation. However, to validate the new methodology, data obtained on the same images with “traditional” methods of analysis should be added for comparison.

- No legends of the figures are provided. It is difficult to understand what the Authors intend to show.

- Method section page 2 “generation and imaging of thrombi”. It is not clear whether the effect of abciximab was tested in parallel with the blood of the same donor used as control. Please add this information. The number of the experiments with and without abciximab should be provided.

- The Author stated that two classes of thrombi were distinguished (page 3, lines 1-6) and are presented in Figure 1. Please describe the figure adding a detailed legend.

- How many images were analysed for each perfusion? Mean ± SD should be added.

- Statistical analysis should be reported.

- The description of the results is difficult to follow. Result section should be rewritten, and more details added.

Reviewer #2: This study presents a machine learning-based method for classification of images of thrombi formed in in vitro flow chambers. Two classes of thrombi are deciphered as the result; a proof-of-concept data set is then presented using known antiplatelet agents.

The machine learning code applied in this platform is used on 18 images from 9 donors as a training set; this is an extremely limited set of images and likely fails to capture the heterogeneity between donors and reproducibility between experiments or within experiments. A complete analysis of the heterogeneity in images within an individual experiments, for the same donor between experiments, and importantly, between donors.

Donor variables including platelet count and sex as a variable should be reported.

Images should be analyzed across a series of timepoints, as this analysis would be unable to distinguish individual thrombi once the platelet coverages becomes confluent. The choice of ‘only high aggregates’ is arbitrary; why when all segments are segmented compared to only high segments, do the y-axis scales change and the larger aggregates appear in the analysis of the high aggregates but are missing in the analysis of all the aggregates.

Comparison to a manual analysis and verification compared to a ground truth is required for validation.

The machine learning code was not developed by this group for this analysis; rather, this study is the demonstration of the use of this code for analyzing platelet thrombus and aggregates. The novelty of this application of analysis tool is not novel and is more suitable for report as a methods paper or used as a method as part of a mechanistic study with a novel finding.

6. PLOS authors have the option to publish the peer review history of their article (what does this mean?). If published, this will include your full peer review and any attached files.

Reviewer #1: No

Reviewer #2: No

---

## [Author Response · Author response to Decision Letter 0]

25 Jan 2024

REVIEWER #1

We have re-evaluated the Ilastik model prior to submission of the revised version of the manuscript and we have decided to train it de novo using better defined criteria for object and pixel classification. These criteria are described in the revised version of the manuscript. We have also modified the method of the model evaluation by introducing confusion matrix and ROC curves. In addition, we have found out that the values presented in the original version were expressed in pixels erroneously described as um. In the present version the values were recalculated.

REVIEWER

- As correctly stated by the Authors the methodology described should not be intended to serve as a replacement of confocal microscopy with three-dimensional reconstruction of images. The new method can help implement image re-analysis as it enables fast calculation. However, to validate the new methodology, data obtained on the same images with “traditional” methods of analysis should be added for comparison.

RESPONSE:

We agree with the suggestion of the reviewer that quantification based on selected class of thrombi should be compared with the traditional approach. The traditional approach is based on quantification of all objects i.e. both separated platelets and thrombi. In our work this approach was performed with the use of a model which segmented all CD41-positive structures. As this idea was not clearly expressed in the original version of our paper, we have amended it in the present version: lines 19-24 page 3

REVIEWER

- No legends of the figures are provided. It is difficult to understand what the Authors intend to show.

RESPONSE:

The figure legends were submitted along with the manuscript as a separate file and were not included in the main manuscript file. In the amended version the legends have been included in the end of the file.

REVIEWER

- Method section page 2 “generation and imaging of thrombi”. It is not clear whether the effect of abciximab was tested in parallel with the blood of the same donor used as control. Please add this information. The number of the experiments with and without abciximab should be provided.

RESPONSE:

The experiments were performed in a pairwise manner. The information has been added in 

Lines 45-46 page 2. The number of repetitions was provided in figure legend.

REVIEWER

- The Author stated that two classes of thrombi were distinguished (page 3, lines 1-6) and are presented in Figure 1. Please describe the figure adding a detailed legend.

RESPONSE

The descritpion of the classes has been provided in the figure legend

REVIEWER

- How many images were analysed for each perfusion? Mean ± SD should be added.

RESPONSE

There were at least 8 images per perfusion as stated in the method section line 1 page 3 and on the figure 7 legend. Therefore, each dot on the Figure 7 represents an average of at least 8 images (technical repetitions). Showing of mean ± SD for each perfusion would hence represent a variability between technical repetitions. Also, these values would be hardly visible when aligned. Thus we decided to omit their presentation. To provide an opportunity for the reviewer and the readers we have published raw data of the experiments in the repository as stated in the Availability of data and materials section.

REVIEWER

- Statistical analysis should be reported.

RESPONSE

Statistical analyses have been added in lines 32-38 page 4

REVIEWER

- The description of the results is difficult to follow. Result section should be rewritten, and more details added.

RESPONSE

We have rewritten the results section

REVIEWER #2: 

We have re-evaluated the Ilastik model prior to submission of the revised version of the manuscript and we have decided to train it de novo using better defined criteria for object and pixel classification. These criteria are described in the revised version of the manuscript. We have also modified the method of the model evaluation by introducing confusion matrix and ROC curves. In addition, we have found out that the values presented in the original version were expressed in pixels erroneously described as um. In the present version the values were recalculated.

REVIEWER

The machine learning code applied in this platform is used on 18 images from 9 donors as a training set; this is an extremely limited set of images and likely fails to capture the heterogeneity between donors and reproducibility between experiments or within experiments. A complete analysis of the heterogeneity in images within an individual experiments, for the same donor between experiments, and importantly, between donors.

RESPONSE:

The approach used in the presented studies is instance segmentation and not image classification. It means that a single element which is being classified here is not an entire image, but a segmented object in the image. Therefore the training set is not actually comprised of 18 images but it consists of 3750 objects of thrombi being segmented in these 18 images. This information has been added to the amended version of the manuscript lines 46-47 page 3 . To ensure a good generalisation of the model the training was performed on samples from different donors. This approach is similar to that used in other publications. In a paper where Ilastik was used for platelet classification, the model was trained on 8 images (Platelets. 2021; 32(1): 54–58.). Authors provided a number of objects in these 8 images: 1732. In another paper (PLoS Comput Biol. 2023 Nov; 19(11): e1010845) authors used five images to train their model. 

A measure of how effective was the training is based on comparision of model outcome with ground truth. This is actually a question of the model fitness, of its ability of generalisation. And this has been evaluated by specific metrics. They have been described in the manuscript.

Heterogeneity of images within an individual images is an intrinsic feature of the microfluidic methods: thrombi can form differently along the channel with more thrombi generated in the inlet than closer to the outlet. This effect is more pronounced in some types of chambers while others are less susceptible. Because of this some researchers prefer to take images only from the defined lengths of the channels to minimize this heterogeneity. But all this considerations are not in the scope of presented paper. The paper is not aimed at refining of the microfluidic chamber assay, but at establishing the methods of analysis of images no matter what kind of microfluidic device was used to generate them. Similarly, variation between donors is a biological effect and not the effect of the image analysis. 

Finally the work presented here was not intended to present a trained model for a production purposes. It is rather a proof of concept work to show, as a title says, that qualitative classification of thrombi prior their quantification could provide better evaluation of the antiplatelet drugs than analysis based on a robust quantification. It is to encourage researchers in the field of thrombosis to employ such a classification-supported in their analysis in place of robust quantification. As it is discussed, the exact method used may differ on several levels: starting from imaging (still or real-time), through staining strategy (other antigens than CD41 can be used for this purpose), and finally the machine learning approach (convolutional neural networks could be employed instead of random forest classifier).

REVIEWER

Donor variables including platelet count and sex as a variable should be reported.

RESPONSE

The data has been added in lines 40-41 page 2.

REVIEWER

Images should be analyzed across a series of timepoints, as this analysis would be unable to distinguish individual thrombi once the platelet coverages becomes confluent. The choice of ‘only high aggregates’ is arbitrary;

RESPONSE:

The timeframe of analysis is usually a choice of researcher. Due to the risk of reaching confluency or, as it is more possible in the case of growing thrombi, due to the risk of occluding the channel, the experiments are usually performed in a period that allows to form separate thrombi. Our study was not aimed at finding a proper way of how long to perfuse the channel, but rather how to analyse the visual data. The choice of high aggregates is in fact arbitrary as we stated it. The reason for this choice has been explained in the paper: the height of aggregates is a crucial parameter as it decides on their occlusive abilities which are clinically relevant. It has been explained in lines 23-24 page 2.

REVIEWER

why when all segments are segmented compared to only high segments, do the y-axis scales change and the larger aggregates appear in the analysis of the high aggregates but are missing in the analysis of all the aggregates.

RESPONSE:

The y-axes present median values. When only high aggregates are segmented, the smallest aggregates are not taken to the analysis which results in a decrease of median. This has been explained in lines 30-36 page 6.

REVIEWER

Comparison to a manual analysis and verification compared to a ground truth is required for validation.

RESPONSE:

We agree with the reviewer’s suggestion that evaluation of the model is substantial. The calculation of Jaccard index (IOU) for both models was performed in the primary version of the manuscript. During revision we have decided to evaluate the model selective towards compacted thrombi with another approach. We have calculated confusion matrices and ROC curves as has been described in the amended version of the manuscript in lines 5-40 page 5.

REVIEWER

The machine learning code was not developed by this group for this analysis; rather, this study is the demonstration of the use of this code for analyzing platelet thrombus and aggregates. The novelty of this application of analysis tool is not novel and is more suitable for report as a methods paper or used as a method as part of a mechanistic study with a novel finding.

RESPONSE:

We fully agree with the reviewer that the paper reports „ the demonstration of the use of this code for analyzing platelet thrombus and aggregates” This was our intention: to encourage researchers who work in the thrombosis field to apply machine learning tools to the analysis by showing that this kind of analysis provides more appropriate results. Therefore we must unfortunately disagree with the statement that “novelty of this application of analysis tool is not novel”. Qualitative evaluation of thrombus images prior to quantitative analysis is rarely performed, and if it is, it has so far been done by "by eye" comparison, as described in the papers we refer to in our manuscript.

---

## [Decision Letter · Decision Letter 1]

7 Feb 2024

Qualitative classification of thrombus images as a way to improve quantitative analysis of thrombus formation in flow chamber assays

PONE-D-23-31983R1

Dear Dr. Przygodzki,

We’re pleased to inform you that your manuscript has been judged scientifically suitable for publication and will be formally accepted for publication once it meets all outstanding technical requirements.

Kind regards,

Giuseppe Remuzzi

Academic Editor

PLOS ONE

Additional Editor Comments (optional):

Reviewers' comments:

Reviewer's Responses to Questions

**Comments to the Author**

1. If the authors have adequately addressed your comments raised in a previous round of review and you feel that this manuscript is now acceptable for publication, you may indicate that here to bypass the “Comments to the Author” section, enter your conflict of interest statement in the “Confidential to Editor” section, and submit your "Accept" recommendation.

Reviewer #1: All comments have been addressed

Reviewer #2: All comments have been addressed

2. Is the manuscript technically sound, and do the data support the conclusions?

Reviewer #1: Yes

Reviewer #2: Yes

3. Has the statistical analysis been performed appropriately and rigorously? 

Reviewer #1: Yes

Reviewer #2: Yes

4. Have the authors made all data underlying the findings in their manuscript fully available?

Reviewer #1: Yes

Reviewer #2: Yes

5. Is the manuscript presented in an intelligible fashion and written in standard English?

Reviewer #1: Yes

Reviewer #2: Yes

6. Review Comments to the Author

Reviewer #1: The revised version of the manuscript Kamola and Przygodzki is improved.

I have no further comments.

Reviewer #2: No further concerns on this study; the authors sufficiently argued their case.

No concerns about dual publication, research ethics, or publication ethics.

7. PLOS authors have the option to publish the peer review history of their article (what does this mean?). If published, this will include your full peer review and any attached files.

Reviewer #1: No

Reviewer #2: No

---

## [Editor Report · Acceptance letter]

1 Mar 2024

PONE-D-23-31983R1 

PLOS ONE

Dear Dr. Przygodzki, 

I'm pleased to inform you that your manuscript has been deemed suitable for publication in PLOS ONE. Congratulations! Your manuscript is now being handed over to our production team.

Kind regards, 

on behalf of

Prof. Giuseppe Remuzzi 

Academic Editor

PLOS ONE